Field evaluation of Mosq-ovitrap, Ovitrap and a CO2-light trap for Aedes albopictus sampling in Shanghai, China

Gao Qiang 1
Cao Hui 1
Fan Jian 1
Zhang Zhendong 1
Jin Shuqing 1
Su Fei 1
Leng Peien lengpeien1380@163.com 2
Xiong Chenglong chenglongxiong@vip.sina.com 3
1 Department of Vector Control, Shanghai Huangpu Center for Disease Control & Prevention , Shanghai , China
2 Department of Vector Control, Shanghai Municipal Center for Disease Control & Prevention , Shanghai , China
3 Department of Epidemiology, School of Public Health, Fudan University , Shanghai , China
Negri Ilaria
Electronic publication date: 2019 Nov 27
Publication date: 2019
Volume: 7
Electronic Location ID: e8031
Received 2019 Jun 13; Accepted 2019 Oct 14
Copyright: ©2019 Gao et al.
Copyright year: 2019
Copyright holder: Gao et al.
License: This is an open access article distributed under the terms of the Creative Commons Attribution License, which permits unrestricted use, distribution, reproduction and adaptation in any medium and for any purpose provided that it is properly attributed. For attribution, the original author(s), title, publication source (PeerJ) and either DOI or URL of the article must be cited.
License URL: https://creativecommons.org/licenses/by/4.0/

Keywords: Ovitrap, CO2-light trap, Mosq-ovitrap, Aedes albopictus, downtown Shanghai

Funding: National Natural Science Foundation of China Youth Fund 81903376 Training Program Foundations for Excellent Youth Medical Talents of Shanghai 2017YQ071 Major Public Health project of Huangpu District HWZFX201805 HKW201624 This work was funded by the National Natural Science Foundation of China Youth Fund (81903376) to Qiang Gao, the Training Program Foundations for Excellent Youth Medical Talents of Shanghai (2017YQ071) to Qiang Gao and the Major Public Health project of Huangpu District (HWZFX201805 & HKW201624) to Qiang Gao. The funders had no role in study design, data collection and analysis, decision to publish, or preparation of the manuscript.

==============================
Background

The Mosq-ovitrap (MOT) is currently used for routine surveillance of container-breeding Aedes in China. However, the effectiveness of monitoring Aedes albopictus using the MOT and other mosquito monitoring methods, such as the Ovitrap (OT) and the CO2-light trap (CLT), have not been extensively compared. Moreover, little is known about the spatial-temporal correlations of eggs with adult Ae. albopictus abundance among these three types of traps.

Methods

Comparative field evaluation of MOT, OT and CLT for Ae. albopictus monitoring was conducted simultaneously at two city parks and three residential neighborhoods in downtown Shanghai for 8 months from April 21 to December 21, 2017.

Results

Significantly more Ae. albopictus eggs were collected from both MOTs and OTs when traps remained in the field for 10 d or 7 d compared with 3 d (MOT: 50.16, 34.15 vs. 12.38 per trap, P < 0.001; OT: 3.98, 2.92 vs. 0.63 per trap, P < 0.001). Egg collections of MOTs were significantly greater than OTs for all three exposure durations (Percent positive: X2 = 72.251, 52.420 and 51.429, P value all < 0.001; egg collections: t = 8.068, 8.517 and 10.021, P value all <0.001). Significant temporal correlations were observed between yields of MOT and CLT in all sampling locations and 3 different MOT exposure durations (correlation coefficient r ranged from 0.439 to 0.850, P values all < 0.05). However, great variation was found in the spatial distributions of Ae. albopictus density between MOT and CLT. MOT considerably underestimated Ae. albopictus abundances in areas with high Ae. albopictus density (>25.56 per day ⋅ trap by CLT).

Conclusion

The MOT was more efficient than the OT in percent positive scores and egg collections of Ae. albopictus. The minimum length of time that MOTs are deployed in the field should not be less than 7 d, as Ae. albopictus collections during this period were much greater than for 3 d of monitoring. MOT considerably underestimated Ae. albopictus abundance in areas with high Aedes albopictus density compared to CLT. In areas with moderate Aedes albopictus densities, MOT results were significantly correlated with CLT catches.

Introduction

Aedes albopictus (Skuse, 1894) (Diptera: Culicidae) is the predominant and the most important arbovirus vector species in the Shanghai region of eastern China (Gao et al., 2014; Zhou et al., 2015). This species is now a top priority of vector control efforts in Shanghai and surrounding areas, especially after the first autochthonous dengue case was recently reported (SHWSJKW, 2017). Vector monitoring is used to estimate Ae. albopictus density and biting rates. However, adult mosquito monitoring approaches are often labor-intensive, unethical (i.e., human landing catches), expensive and difficult to implement on a large scale (i.e., BG sentinel traps and CO2-light traps) (Manica et al., 2017). Artificial traps for Aedes (Stegomyia) egg collection (i.e., Ovitrap, Mosq-ovitrap) are relatively easy and inexpensive to construct. They are also easier to implement in the field because they are portable and do not require electricity or CO2. Egg traps can detect the presence of gravid Aedes females, even at low population densities. This makes them good alternatives for Aedes monitoring (Silver, 2008). If statistical correlations between egg-collection data and adult population density can be demonstrated, then the workload and cost for adult Aedes monitoring could be reduced.

The Ovitrap (OT) was initially developed during the first 3 years of the USA Ae. aegypti Eradication Program. It was used to detect the presence of Ae. aegypti (Fay & Eliason, 1966; Fay & Perry, 1965; Jakob & Bevier, 1969). The OT is now a common Aedes monitoring approach in many regions, including North and South America, Europe and Asia, due to its high sensitivity and low implementation costs (ECDC, 2012; Mogi et al., 1988; Reiter, Amador & Colon, 1991). Mattia (Manica et al., 2017) demonstrated the possibility of predicting the biting rate of Ae. albopictus based on OT egg-collection data. However, the reliability of OT data for estimating adult population abundance has been debated (Focks, 2003). The Mosq-ovitrap (MOT), based on the OT, was developed by Lin et al. (2005). It is a safe, efficient and economical method for conducting Aedes surveillance. The trap design makes it difficult for captured gravid Aedes females to escape, so MOT can quantitatively measure both adult females and their egg production.

The MOT was developed, and is mainly used, in China. Its field sampling efficiency, in comparison to other traps, is poorly known. Lin et al. (2006b) demonstrated positive correlations between MOT and OT for some macro indices, such as the positive index (percentage of traps containing adult mosquitoes or eggs), but detailed indices, including egg counts collected by the two traps, were not compared. The MOT has been widely used for routine Ae. albopictus monitoring in China (Li et al., 2016), and the mosq-oviposition positive index (MOI) lower than 5 has been used as the threshold for Ae. albopictus population density being less than the level requiring control treatment. The sampling efficiency of the MOT has not been extensively compared with other adult mosquito monitoring methods, such as human landing catch or CO2-light trap (CLT). (Li et al., 2016) compared MOT with the BG-trap and the CDC light trap, but, due to the small number of captures, the MOT data were not analyzed. Therefore, little is known about the spatial–temporal correlations of eggs or adult mosquito catches between MOT and adult mosquito traps. The surveillance duration of MOT in China is typically 3 d compared to the widely accepted 7 d interval for field surveillance using CDC ovitraps (Reiter, Amador & Colon, 1991). It is not known if the field surveillance time for MOT should be extended. It is also unknown if the low mosquito capture efficiency of MOT in the Li et al. (2016) study was caused by the 3 d exposure duration.

To address these questions, we conducted a comparative field evaluation of MOT, OT and CLT for Ae. albopictus monitoring in urban areas of Shanghai. We determined the best sampling duration for MOT to maximize Ae. albopictus catches and then conducted correlation analysis and efficiency evaluations between MOT and OT. For adult mosquito density, spatial–temporal correlations of eggs or adult mosquito catches between MOT and adult mosquito trap CLT were made. The advantages and disadvantages of MOTs used as surveillance tools for Ae. albopictus population level were then described.

Materials and Methods

Study area

The study was conducted in 5 downtown areas of Shanghai, China (31°13′N, 121°27′E, el 3.5 m). The areas consisted of 2 city parks and 3 residential neighborhoods (Fig. 1). A total of 8 field sites were used for comparisons among the CLT, MOT and OT trap. Detailed geographical coordinates of the 8 sites were described in a previous report (Gao et al., 2016). We placed one CLT (total = 8 CLT) and 2 or 3 pairs (total = 19 pairs) of MOTs and OTs in each site for comparison. To reduce direct competition among the traps, all traps within one site were separated by an average distance of 20 m.

Traps tested

The MOT (Tianpai, Kaiqi Co. Ltd, Shanghai, China) used was described by Lin (Lin et al., 2005). It consists of a transparent cylindrical plastic jar (10 cm high ×7 cm diam.) with a concave bottom (about two cm inward) and a black top cover with 3 conical holes (one cm diam.) (Fig. 2). A white circular filter paper (7.5 cm in diameter) purchased from Aoke Co. Ltd (Taizhou, China) is placed inside the bottom of the jar as an oviposition substrate, and 20 ml of dechlorinated tap water is added to the jar to keep the filter moist but not submerged. The conical holes in the cover are designed for easy entry but difficult exit for the mosquitoes. Consistent with the observations of Lin (Lin et al., 2005), we also noted that captured gravid mosquitoes typically do not escape (escape rate was <10%, 15/162 = 9.26%) (Q Gao, 2016, unpublished data), and they will lay eggs on the paper substrate inside the trap. Therefore, this trap can provide a quantitative measure of the number of adult female mosquitoes and egg production. MOTs were placed on the ground in relatively secluded locations near vegetation and protected from rain.

Figure 1 Trap locations for mosquito monitoring comparisons of MOT, OT and CLT.

Map data 2019 OpenStreetMap.

Figure 2 Mosq-ovitrap (A and C) and Ovitrap (B and D) used in this study; (E): a three-dimensional diagram of MOT; (F): a cross-sectional diagram of MOT; (G): a cross-sectional diagram of MOT with water and filter paper.

The OT used in this study was purchased from Tianpai, Kaiqi Co. Ltd. (Shanghai, China). It consists of a black tinted plastic jar with straight, slightly tapered, sides (Fig. 2). The OT is approximately 10 cm high with an opening diameter of seven cm. A plastic paddle is affixed to inside wall of the jar, with an overflow hole three cm from the top. About 118 ml of dechlorinated water was added to the jar, and a white filter paper strip (12 × two cm) was used as an oviposition substrate.

The CLT used in this study was structurally similar to the Center for Disease Control and Prevention light-traps (Bat King Co. Ltd, Shanghai, China). The trap consists of an AC/battery-powered fan, a trap bag and an ultraviolet lamp (Fig. 3). We also used a compressed CO2 gas cylinder (5 kg) and a trap bait, produced by Bat King, that simulated human body scent. The CO2 gas cylinder and trap bait were replaced every 60 d.

Figure 3 Field mosquito collection in three traps.

(A, B, C) adult mosquitoes and eggs trapped by Mosq-ovitraps; (D, E) eggs trapped by ovitraps; (F) adult mosquitoes trapped by CO2-light traps.

Mosquito sampling

Mosquito sampling was conducted during 8 months from April 21 to December 21, 2017. MOTs and OTs were left in the field for 10 d. The number of adult mosquitoes trapped and the number of eggs produced and trap conditions (i.e., tipped, dried, missing, flooded or broken) were recorded after 3, 7 and 10 d, respectively, for each sample interval. Collections were conducted between 0900 and 1200 h, which is the daily time of lowest mosquito activity (Haddow & Gillett, 1957; Reiter, Amador & Colon, 1991). Positive MOT or OT was defined as those containing at least one adult, egg or larva in the trap. CLT sampling of adult mosquitoes was conducted for 24 h every 10 d during the 8 months. The trapped mosquitoes were collected and stored at −80 °C.

Measure of field mosquito collection yields

MOTs collect both adult mosquitoes and eggs. Since the collection yields were checked 3 times (after 3, 7 and 10 d), the number of mosquitoes and eggs trapped by MOTs could be field counted when mosquito abundance was relatively low. However, it was difficult to conduct accurate counts when there were large numbers of eggs. In this case, high definition photos were taken of the eggs in the traps, and the egg number was determined later by magnifying the photos. This method was also applied to egg collection measurement for the OTs. Mosquitoes collected by CLT were taken to the laboratory for identification and processing.

We used three measurements of mosquito trap sampling: percent positive, egg collection and mosquito density. “Percent positive” is defined as the percentage of positive MOT or OT in the total traps used for sampling (a positive trap has at least one egg, adult or larva). “Egg collection” refers to the number of eggs collected by traps, and “Mosquito density” refers to the density of trapped adults.

Mosquito processing

Adult mosquitoes trapped by CLTs or MOTs were killed by freezing and then counted and identified using taxonomic keys (Becker et al., 2003). Eggs or eclosed larvae collected by MOTs and OTs after 10 d were taken back to the laboratory. For species identification, we randomly selected 20% of the trapped eggs or larvae and then hatched and reared these until adult emergence. Adults were identified and the results were extrapolated to the rest of the unhatched eggs.

Statistics

Data were analyzed using SPSS ver. 11.5 (SPSS, Inc., Chicago, IL, USA) statistical software package. There were different measurements for mosquito collection in this study, including percent positive (%), egg collection (per trap) and mosquito density (per trap). The variance of percent positive (%) among three exposure durations was compared by using the Kurskal-Wallis H test, and variances in the different trap types were compared by Pearson chi-square test. For quantitative data of “egg collection” or “mosquito density,” independent t test or one-way ANOVA was used for analysis and Bonferroni test for pairwise comparisons. Correlation analysis was used for trend comparison between the different trap types. P < 0.05 represented a significant difference.

Results

Mosquito population structure based on captures of the three traps

A total of 21,919 and 1,740 eggs were collected by the MOT and OT, respectively. A randomized 20% of these eggs were reared to adults, and all were Ae. albopictus. A total of 370 adult Ae. albopictus (♀: ♂ = 355:15) and 3 adult female Culex pipiens complex were collected by the MOT; the Cx. pipiens were only trapped in city park II in mid-November. Four species were collected by CLTs, including 4,858 Ae. albopictus (♀: ♂ = 3,007:1,851), 475 Cx. pipiens complex (♀: ♂ = 314:161), 338 Culex tritaeniorhynchus ( ♀: ♂ = 335:3), and 9 ♀ Anopheles sinesis. It is not within the scope of this paper to present results for other mosquito species from the 3 traps; and so the results are limited to Ae. albopictus.

Comparison between MOT and OT

For the two measurements of percent positive and egg collections (Table S1), MOT yields were significantly greater than OTs for all of the three exposure durations (percent positive: X 2 = 72.251, 52.420 and 51.429, P values all <0.001; egg collection: t = 8.068, 8.517 and 10.021, P values all <0.001) (Tables 1 and 2).

For both MOT and OT, the percent positive and egg collections both significantly increased with increased exposure duration (Kruskal-Wallis H test for percent positive: X2 = 7.050, P = 0.029 of MOT, X2 = 17.678, P < 0.001 of OT; one-way ANOVA for egg collection: F = 4.854, P = 0.012 of MOT, F = 7.632, P = 0.001 of OT). At 7 d, the percent positive was significantly greater than the percent at 3 d (MOT: 42.33 vs. 28.15%, X2 = 19.272, P < 0.001; OT: 19.68 vs. 6.41%, X2 = 33.935, P < 0.001) (Fig. 4A). This was also true for the yield of egg collections (MOT: 34.15 vs. 12.38 per trap, P < 0.001; OT: 2.92 vs 0.63 per trap, P < 0.001) (Fig. 4B). A lesser increase of percent positive was observed at 10 d exposure compared with 7 d for both MOT and OT (MOT: 44.85 vs. 42.33%, X 2 = 0.563, P = 0.453; OT: 21.97 vs. 19.68%, X 2 = 0.694, P = 0.405). Egg collections were significantly greater at 10 d compared with 7 d for MOT (P = 0.011), but not significant for OT (P = 0.229) (Table 2, Figs. 4A and 4B).

Table 1 Comparisons of percent positive for traps with different types and different exposure durations.

(a) MOT of 3 d; (b) MOT of 7 d; (c) MOT of 10 d; (d) OT of 3 d; (e) OT of 7 d; (f) OT of 10 d.

	Positive
index I	Positive
index II	X2 value
(chi-square)	P value	
MOT vs. OT with the same exposure durations	a: 28.15%	d: 6.41%	72.251	<0.001	
b: 42.33%	e: 19.68%	52.420	<0.001	
	c: 44.85%	f: 21.97%	51.429	<0.001	
MOT: Different exposure durations	a: 28.15%	b: 42.33%	19.272	<0.001	
a: 28.15%	c: 44.85%	26.307	<0.001	
b: 42.33%	c: 44.85%	0.563	0.453	
OT: Different exposure durations	d: 6.41%	e: 19.68%	33.935	<0.001	
d: 6.41%	f: 21.97%	43.456	<0.001	
e: 19.68%	f: 21.97%	0.694	0.405	

Table 2 Comparisons of mean egg collections for traps with different types and different exposure durations.

(A) MOT of 3 d; (B) MOT of 7 d; (C) MOT of 10 d; (D) OT of 3 d; (E) OT of 7 d; (F) OT of 10 d. p >.

	Mean egg collection I	Mean egg collection II	t / F	P value	
MOT vs OT with the same exposure durations	A: 12.38	D: 0.63	t = 8.068	<0.001	
B: 34.15	E: 2.92	t = 8.517	<0.001	
	C: 50.16	F: 3.98	t = 10.021	<0.001	
MOT: Different exposure durations	A: 12.38	B: 34.15	F = 28.911	<0.001	
A: 12.38	C: 50.16		<0.001	
B: 34.15	C: 50.16		0.011	
OT: Different exposure durations	D: 0.63	E: 2.92	F = 25.405	<0.001	
D: 0.63	F: 3.98		<0.001	
E: 2.92	F: 3.98		0.229	

Figure 4 Comparation of mosquito collections between MOT and OT.

(A and B) Comparison of mosquito collection indices between MOT and OT with different exposure durations. (C and D) Seasonal dynamics of water evaporation rate (%) of MOT and OT with different exposure durations.

Although the temporal trends of mosquito yields between MOT and OT were significantly correlated (Pearson’s correlation coefficient r = 0.413, P = 0.023), there was a difference between these traps. MOT monitoring indicated that the Ae. albopictus population peaked in July and August (Figs. 5A, 5C and 5E). However, the largest egg collections for OT appeared in June, and there was an obvious decline in July (Figs. 5B and 5D).

Figure 5 Seasonal dynamics of percent positive (%), egg collections and mosquito density of MOT and OT among three exposure durations.

Comparison between MOT and CLT

Correlation analysis showed that there were strong temporal correlations between yields of MOT and CLT at all five sampling fields and the three different MOT exposure durations (Table 3 and Fig. 6). Both traps indicated that Ae. albopictus populations peaked in July and started to decline in August (Table S2). CLT collected adult mosquitoes, and its efficiency in Ae. albopictus sampling was significantly higher than MOT (26.41 vs. 0.85 per trap, t =  − 3.995, P = 0.002).

Table 3 Correlation analysis between Ae. albopictus yields (adults and eggs) of MOT and CLT with different environmental locations.

Surveillance durations of MOT	Sites	MOT	CLT	r (pearson correlation coefficient)	P value	
		indicators I for correlation analysis	indicators II for correlation analysis			
After 3 d	City park I	adult Ae. albopictus	adult Ae. albopictus	0.656	0.001	
		albopictus eggs	adult Ae. albopictus	0.669	0.001	
	City park II	adult Ae. albopictus	adult Ae. albopictus	0.596	0.003	
		albopictus eggs	adult Ae. albopictus	0.715	<0.001	
	3 Residential neighborhoods	adult Ae. albopictus	adult Ae. albopictus	0.492	0.02	
		albopictus eggs	adult Ae. albopictus	0.801	<0.001	
	Sum	adult Ae. albopictus	adult Ae. albopictus	0.582	0.005	
		albopictus eggs	adult Ae. albopictus	0.610	0.003	
After 7 d	City park I	adult Ae. albopictus	adult Ae. albopictus	0.699	<0.001	
		albopictus eggs	adult Ae. albopictus	0.650	0.001	
	City park II	adult Ae. albopictus	adult Ae. albopictus	0.583	0.004	
		albopictus eggs	adult Ae. albopictus	0.439	0.041	
	3 Residential neighborhoods	adult Ae. albopictus	adult Ae. albopictus	0.616	0.002	
		albopictus eggs	adult Ae. albopictus	0.751	<0.001	
	Sum	adult Ae. albopictus	adult Ae. albopictus	0.635	0.001	
		albopictus eggs	adult Ae. albopictus	0.545	0.009	
After 10 d	City park I	adult Ae. albopictus	adult Ae. albopictus	0.797	<0.001	
		albopictus eggs	adult Ae. albopictus	0.850	<0.001	
	City park II	adult Ae. albopictus	adult Ae. albopictus	0.645	0.001	
		albopictus eggs	adult Ae. albopictus	0.610	0.003	
	Residential neighborhood	adult Ae. albopictus	adult Ae. albopictus	0.672	0.001	
		albopictus eggs	adult Ae. albopictus	0.836	<0.001	
	Sum	adult Ae. albopictus	adult Ae. albopictus	0.789	<0.001	
		albopictus eggs	adult Ae. albopictus	0.780	<0.001	

Figure 6 Correlation analysis between Ae. albopictus yields of MOT (10 d exposure duration) and CLT.

(A) adult Ae. albopictus of CLT and adult Ae. albopictus of MOT; (B) adult Ae. albopictus of CLT and Ae. albopictus egg collection of MOT.

CLT samples showed that Ae. albopictus were significantly more abundant in residential neighborhoods than in city parks (58.71 vs. 7.22 per day ⋅ trap, t =  − 3.203, P = 0.004). An opposite pattern was found for MOT (adults: 0.39 vs. 1.25 per trap, t =  − 3.203, P = 0.004; eggs: 19.89 vs. 79.18 per trap, t = 3.836, P < 0.001) (Table 4, Figs. 7A–7C). No significant correlations were observed among locations between the indices of CLT and MOT (correlation coefficient r = −0.124, P = 0.774) (Figs. 8A and 8B). However, for the 5 groups of traps located in city parks with moderate Ae. albopictus density (<13.77 per day ⋅ trap by CLT), a significant correlation was found between sampling yields of CLT and MOT (correlation coefficient r = 0.976, P = 0.004) (Figs. 8C and 8D).

Table 4 Comparison of Ae. albopictus yields (adults and eggs) of MOT and CLT in different environmental locations.

Sampling fields	CLT	MOT	
	Trap locations	Adult density (per day⋅ trap)	Trap locationsa	Adult density (per trap)	Egg density (per trap)	
City park I	1	1.43	1 & 2	0.39	17.17	
2	1.30	3 & 4	0.80	48.24	
3	15.87	5 & 6	2.72	145.39	
City park II	4	3.17	7 & 8	0.80	53.57	
5	13.30	9 & 10	1.59	122.61	
Residential neighborhoods	6	153.57	11, 12 & 13	0.65	32.14	
7	13.57	14, 15 & 16	0.45	24.68	
8	9.00	17, 18 & 19	0.06	2.86	
Notes.

a locations of MOTs corresponding to CLTs, among the 8 locations, each CLT (total = 8 traps) was accompanied by 2 or 3 MOTs (total = 19 traps).

Figure 7 Comparison of Ae. albopictus yields (adults and eggs) of MOT and CLT in different environmental locations.

(A) Adult Ae. albopictus of MOT; (B) Ae. albopictus eggs of MOT; (C) adult Ae. albopictus of CLT.

Figure 8 Correlation analysis between Ae. albopictus yields (adults and eggs) of MOT and CLT in areas with different mosquito densities.

(A and B) No correlations between CLT and MOT. (C and D) Significant correlations between CLT and MOT in areas with moderate Ae. albopictus density.

Discussion

In this study, more Ae. albopictus eggs were collected from MOTs than OTs. We found that the minimum length of time that MOTs are exposed in the field should be at least 7 d. Significant temporal correlations were observed between sampling measurements of MOT and CLT, but there was substantial variation in the spatial distribution of Ae. albopictus density between MOT and CLT. MOT underestimated Ae. albopictus abundances in areas with high Ae. albopictus density compared to CLT.

Before the development of the MOT, the OT was regarded as an efficient and sensitive method for detecting the presence/absence of dengue vectors, even at low population densities (Chadee & Corbet, 1987; Evans & Bevier, 1969). MOT was designed by Lin (Lin et al., 2005), who compared MOT and OT as tools for Ae. albopictus monitoring. They demonstrated a positive correlation between the captures made by the two traps. However, MOT was somewhat less productive compared to OT (Lin et al., 2006a). We also observed positive correlations between the indices of MOTs and OTs, but we found that the MOT was more efficient than OTs for sampling Ae. albopictus eggs. Chadee (Chadee & Corbet, 1987) also reported that the number of traps receiving eggs and the number of eggs per positive trap were usually lower for OTs. Therefore, modifications were developed for trap enhancement. Hay infusion, as attractants, and hard fiberboard paddles, as an oviposition substrate, increased the eggs yields for OTs (Reiter, Amador & Colon, 1991; Ritchie, 2001). However, these modifications were not used in this study. To match the conditions of MOTs and strengthen comparability, dechlorinated tap water and filter paper oviposition substrate were used for both MOTs and OTs. Tap water and filter paper may not be optimum attractants for use in OTs. Another important reason may be attributed to the special structural design of MOTs with easy entrance access but difficult egress. Gravid Ae. albopictus females may hover around MOTs or OTs and hesitate before laying eggs in these artificial containers. A percentage of the females contacting OTs does not lay eggs in the trap, perhaps due to lack of suitable oviposition stimulants. A portion of females trapped by MOTs had no choice but to lay eggs inside the MOTs since they were unable to exit. This greatly increased the egg collections of the MOTs. So, under the same conditions, MOTs typically collected more eggs than OTs.

MOTs are used for mosquito surveillance in China, and the currently recommended inspection duration is 3 d. This duration was based on field tests (Lin et al., 2006a). They found that mosquito yields increased significantly with increased exposure duration, but durations longer than 3 d increased the mortality of the mosquitoes in the MOTs. We suggest that collection of live mosquitoes is less important than precise surveillance outcomes. MOTs were designed to attract oviposition-seeking females (Li et al., 2016), and a longer duration attracts more females and allows the captured gravid females enough time to lay eggs, which increases the monitoring sensitivity. The rapid increase of percent positive and egg collections during 7 d indicates that the appropriate monitoring duration for MOTs surveillance should be no less than 7 d. Moreover, a widely accepted scheme using CDC ovitraps in the field for 7 d of surveillance provides support for this suggestion (Reiter, Amador & Colon, 1991). Our tests also showed that high evaporation rates can occur during the warm summer season (July and August in Shanghai). Therefore, water needs to be replenished during the 7 monitoring days when temperatures are high.

Ovitrap data may help provide estimates of adult Aedes populations in some case. Mattia (Manica et al., 2017) showed that it is possible to predict the abundance of adult Ae. albopictus females based on egg collections. A significant positive relationship was found between the mean numbers of Ae. albopictus females collected by human landing catch and Ae. albopictus eggs collected by OTs. However, this type of positive relationship between Ae. albopictus eggs and adult females was not fully reproduced in the present study using MOT. As a modified ovitrap, MOT correlated with CLT in areas with moderate Ae. albopictus densities (<13.77 per day ⋅ trap by CLT) but underestimated Ae. albopictus abundances in areas with high densities (>25.56 per day ⋅ trap by CLT) compared with CLT. The latter phenomenon was consistent with the suggestion by Focks (2003) that ovitrap data cannot be reliably used to estimate the differences in adult population abundance. This result was predictable. Ae. albopictus has strong oviposition preference for particular sites, and this preference is governed by the combined effects of many physical and chemical stimuli that act over a range of distances (Silver, 2008). Thavara et al. (1989) demonstrated that Ae. albopictus prefers to lay eggs in containers holding conditioned water that has been “aged” outside for an extended period together with harboring the immature stages of this species. MOT is not a preferred oviposition site for Ae. albopictus compared to natural containers with conditioned water. Areas with relatively low Ae. albopictus density may lack natural breeding sites, and, in these areas, it is reasonable that gravid Ae. albopictus females would lay eggs in the artificial MOTs since there are few preferred choices. In areas with high Ae. albopictus density, it may be easy for them to find preferred oviposition sites, and MOTs in these areas would be less-preferred oviposition choices.

In this study, MOTs collected mostly Ae. albopictus, and few other species were attracted. This may have resulted because of a) the relatively limited mosquito species composition in urban Shanghai, and b) the different oviposition preferences of different mosquito species. In urban Shanghai, the predominant mosquito species are Ae. albopictus and Cx. pipiens complex. Other species like Cx. tritaeniorhynchus and An. sinesis are rare (Gao et al., 2014). Due to its design, the MOT containing water and filter paper was aimed to mainly attract gravid female mosquitoes and allow them to ovoposit (Li et al., 2016). Among the common mosquito species in downtown Shanghai, Ae. albopictus prefers to lay eggs in container water, while Cx. pipiens prefers to lay eggs in waste or polluted water. Cx. tritaeniorhynchus and An. Sinesis both prefer large water bodies like paddy fields or irrigation ditches, which are more common in rural areas. So the physical features of water and the MOT design are unsuitable for the other mosquito species of urban Shanghai.

In the field test, more data losses of OTs occurred than MOTs. The causes of data losses were primarily due to dried, tipped, and missing container and animal interference. OTs have an upper part completely exposed to the air making it possible for small animals (i.e., dogs and cats) to drink the water in the trap. During the hot summer, water inside the trap tends to evaporate rapidly. The open-design of OTs increases water evaporation compared with the semi-open design of MOT, and there is greater drying caused by water evaporation for OT in July and August compared with MOT (Figs. 4C and 4D). This may help explain the considerable variance of temporal trends of mosquito yields between MOT and OT. Flooding was a major cause for data losses for MOTs without overhead shelter. In rainy periods, excessive rain can flow into the traps and submerge the filter paper. We suggest that a drainage hole should be drilled in the middle of the MOT’s concave-bottom to reduce flooding problems. We also found spiders and geckos entering the trap and consuming mosquitoes and the eggs collected by the traps. However, snails or roaches, which also may eat or dislodge eggs, were not found in this study (Kloter, Bowman & Carroll, 1983).

MOTs have certain shortcomings. Ovitrap data cannot be reliably used to estimate the differences in adult Ae. albopictus abundance, and MOT data were unable to predict differences of Ae. albopictus population abundance among different locations. In China, percent positive of MOT < 5% or MOT positive index <5 (MOI < 5) has been used as a threshold to characterize Ae. albopictus density as sufficiently low as to not require control. This criterion was established based on the Breteau Index (BI: number of positive small water bodies per 100 houses inspected), which represents the positive percentage of natural breeding sites, whose threshold was also <5 (BI <5). However, the percent of positive MOTs cannot estimate or be equal to the Breteau Index, since these two indices have a competitive relationship. Even though field tests in Guangdong Province suggest that the positive index of MOTs (MOI, number of positive traps per l00 MOTs) was almost equal to Breteau Index (Lin et al., 2006c), we believe that this equivalency is probably not widely applicable. A field test in Shanghai demonstrated that significant differences exist between MOI and BI (Pan et al., 2016). Based on these arguments, we suggest that MOI cannot replace or be equal to BI. Both of these indices should be used for assessment of Ae. albopictus oviposition or breeding status.

The results of this study had the following shortcomings: (a) The maintenance and servicing of MOTs or OTs was labor intensive as was egg counting and species identification. CLT collection performance was also expensive. We therefore established only 8 sites for comparisons among MOTs, OTs and CLTs. However, the relatively small sample sizes provided useful initial results. More convincing evidence to support our tentative conclusion will require comprehensive, large-scale field comparisons; (b) To reduce direct competition, all traps within each site were separated by an average distance of 20 m. However, this did not completely eliminate competition. A latin-square design would have been superior, but the CLT used required a DC power supply, which limited the possible CLT locations. Despites these shortcomings, continuous sampling over 8 months provided useful information about MOTs, which can be of help in improving designs for future Ae. albopictus surveillance.

Conclusions

MOT is more efficient than OT in measurements of percent positive and egg collections of Ae. albopictus. The minimum length of time that MOTs are deployed in the field should be at least 7 d. Strong temporal correlations were observed between sampling measurements of MOT and CLT, but there was substantial variation in the spatial distribution of Ae. albopictus density measured by MOT and CLT. MOT underestimated Ae. albopictus abundances in areas with high Ae. albopictus density (>25.56 per day ⋅ trap by CLT). In areas with moderate Ae. albopictus densities (<13.77 per day ⋅ trap by CLT), MOT data were better correlated with CLT data.

Supplemental Information

File S1 Raw data of MOT, OT and CLT in different sampling date and locations

Click here for additional data file.

Table S1 Percent positive (%) and egg collections of MOT and OT of different exposure durations in different locations

a,b,c,d,e,f positive index corresponding to comparative parameters of Table 2. A,B,C,D,E,F index of average egg collections corresponding to comparative parameters of Table 3.

Click here for additional data file.

Table S2 Ae. albopictus yields for MOT (after 10 d exposure) and CLT with different environmental locations

a Mosquito yields of MOTs after 10 d exposure duration were used for comparison, which represent the maximum collections for one interval in this study.

Click here for additional data file.

We thank LetPub for providing linguistic assistance during manuscript preparation.

Additional Information and Declarations

Competing Interests

Author Contributions

Field Study Permissions

Data Availability

The authors declare there are no competing interests.

Qiang Gao conceived and designed the experiments, performed the experiments, analyzed the data, contributed reagents/materials/analysis tools, prepared figures and/or tables, authored or reviewed drafts of the paper, approved the final draft.

Hui Cao performed the experiments, contributed reagents/materials/analysis tools, authored or reviewed drafts of the paper, approved the final draft.

Jian Fan, Zhendong Zhang, Shuqing Jin and Fei Su performed the experiments, authored or reviewed drafts of the paper, approved the final draft.

Peien Leng and Chenglong Xiong conceived and designed the experiments, authored or reviewed drafts of the paper, approved the final draft.

The following information was supplied relating to field study approvals (i.e., approving body and any reference numbers):

The field work was approved by Huangpu Center of Disease Control and Prevention, and the approval number is HPCDC-SY20170302.

The following information was supplied regarding data availability:

The raw measurements are available in the Supplementary Files.

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
