# Peer review of "Field evaluation of Mosq-ovitrap, Ovitrap and a CO2-light trap for Aedes albopictus sampling in Shanghai, China"

_PeerJ, doi:10.7717/peerj.8031_

## Round 0.1 · original submission · Major Revisions

In my opinion, the manuscript needs major revisions prior to publication.
First of all, the authors should carefully revise English language.
Second, the authors need to provide more details on the experimental methods, including the statistical analysis. For example, authors should explain how they avoided competition between Mosq-ovitrap, Ovitrap and CO2-light traps, and should provide details on “mosquito sampling” for CO2-light trap; on the number of eggs/hatched larvae collected from MOT and OT for further identification of adults; on the statistics applied to data; on the taxonomic keys used in the study for species recognition.
Statistical analysis and discussion need also to be improved.
Some statements in the text should be better supported. As an example, the sentence “We have observed that captured gravid mosquitoes typically do not escape and they will lay eggs on the paper substrate inside the trap” should be supported by data or literature.
Some data should be presented in a clearer manner: for example, please provide a better explanation of the comparison between “mean egg collection in MOT” and “mosquito density in CLT”.

Authors should be more explicit if only the species Aedes albopictus has been captured by the traps or not; in the latter case, authors should discuss why no other mosquito species have been captured.
Finally, the manuscript could also be improved by merging figure 2 and 3 into a single figure, and by moving Table 1 and 5 into supplementary materials. Table 4 should be revised also considering concerns on the comparison made between eggs and adults captured by MOT and CLT traps.

·

Basic reporting

The authors compared Ae. albopictus sampling with MOTs, OTs and CLTs in Shanghai, China. They present that more eggs were collected in MOTs compared to OTs and that the minimum time MOTs should be deployed is 7 days.
The manuscript is of very good quality – clear and prof. English – and presents relevant Figures and Tables.

Experimental design

The research question is well defined and methods described.
It is clearly mentioned and discussed that OTs were slightly modified.

Validity of the findings

The authors clearly describe the advantages of MOTs compared to OTs for albopictus egg sampling. Moreover adult sampling is compared of MOTs and CLTs.
Conclusions are well stated.

Additional comments

However some modifications are recommended
1) Sometimes the authors only mention Aedes. Do you mean albopictus or also other Aedes species. Please change to albopictus if you only mean Ae. albopictus
2) entire manuscript – change to CO2 (2 smaller; e.g. line 58)
3) Mosquito processing: Which keys were used for specification?
4) Line 243: change to MOTs (Li et al… - Space – also at Line 255
5) Figure 9: Legend: When mentioning Aedes - only albopictus included?

·

Basic reporting

# please use more scientific phrasing  e.g. do not use dangerous (line 52), but most important

# improvement English need significant improvement for the complete manuscript  “egg hatch” (line 164), “use generalized linear mixed model for analysis“ (lines 175-175), “percent positive was“ (line 182), etc.

Experimental design

# lines 75-89: Please highlight the lack of knowledge and the aim of this study to tackle this lack of knowledge. In the moment this is a little bit lost in the presentation of the literature.

# line 95: What do you mean by “one group”? Did you just put all traps in one place? Why did you not use a latin-square approach to account for small-scaled differences between trapping sites and to avoid direct competition between the traps?

# lines 107-: How did you observed that gravid mosquitoes typically do not escape? Please present data and/or publication.

# lines 110-: “Traps were also…”  at the end of the “Traps Tested”-section

# lines 125-131: How were the sampling with the CLT conducted? How often? Time? Etc.

# lines 133-140: I thought you reared the mosquitoes to adults (see line 150)?

# line 149: Which taxonomic keys?

# line 150: How many eggs were selected?

# lines 154-160: The statistics are unclear. Which kind of GLMM did you use (poisson, etc.). What is the random effect? What did you mean by “preliminary”? Why do you use different tests?

# lines 174-176: This belongs to the methods section

Validity of the findings

# lines 181-188: You should not use the phrase “increasing” as you do not observe this process, but comparing different groups (3d, 7d, 10d)

# lines 181-188: You need a p-value correction (e.g. Bonferroni) for multiple tests

# lines 189-191: Please analyse the correlation between the MOT and OT samples over time

# lines 191-192: Discussion belongs to the discussion section

# lines 196-197: I do not understand how and why can compare eggs and adults?

# lines 198: test statistics of the Pearson correlation?

# tables/figures: 9 Figures and 6 Tables are to many and make it difficult the capture the important results. Please think about to move some of the tables/figures to the supplement (e.g. Table 1 and Table 4)

Additional comments

# lines 52-63: please explain more precise why ovitraps are much easier to implement (e.g. no electricity, no CO2, etc.)

# lines 61-70: please give a more comprehensive review on studies showing the correlation or the opposite

# 240-242: I do not understand this sentence. What do you mean by “secondary index?”

# lines 269-280: What are the consequences from this?

# lines 292-296: The paragraph appears a little bit hasty. Please give a comprehensive discussion of the potential problems of the here presented study (e.g. why did you not use a latin-square approach)

# lines 299: “more productive”  do you mean efficient?

---

## Round 0.2 · accepted · Accept

In my opinion, the current version of the manuscript has been significantly improved. Authors addressed the requests made by reviewers, overcoming major problems. In particular, authors provided a revised map of the study area with more accurate locations of sampling sites. Shortcomings, including possible competition between the different mosquito traps, have been satisfactorily discussed.

Figures and tables have been also significantly improved. Other important details have been provided in the materials and method section which better clarified how data were collected for example from CO2-light traps or for species identification following eggs/larvae collection from mosquito ovitraps.

Statistics has been revised (e.g., authors chose to eliminate generalized linear mixed model (GLMM)) and properly discussed.

Some unclear points have been deleted or edited (e.g., why mostly Aedes albopictus and few other species were attracted by traps), improving the clarity and accuracy of the article. Authors also added missing information e.g., taxonomic keys used in the study.

·

Basic reporting

The authors signiciantly improved their manuscript and responded to all of my comments,

Experimental design

The authors signiciantly improved their manuscript and responded to all of my comments,

Validity of the findings

The authors signiciantly improved their manuscript and responded to all of my comments,

Additional comments

The authors signiciantly improved their manuscript and responded to all of my comments,